# Hepatitis E Virus Infection in Patients with Chronic Inflammatory Bowel Disease Treated with Immunosuppressive Therapy

**DOI:** 10.3390/pathogens12020332

**Published:** 2023-02-15

**Authors:** Ilias Kounis, Christophe Renou, Stephane Nahon, Frederic Heluwaert, Gilles Macaigne, Morgane Amil, Stephane Talom, Benedicte Lambare, Claire Charpignon, Thierry Paupard, Monica Stetiu, Marie Pierre Ripault, Armand Yamaga, Florent Ehrhard, Franck Audemar, Maria Carmen Ortiz Correro, David Zanditenas, Florence Skinazi, Helene Agostini, Audrey Coilly, Anne Marie Roque-Afonso

**Affiliations:** 1Centre Hépato-Biliaire, AP-HP Hôpital Paul-Brousse, 94800 Villejuif, France; 2Inserm, UMR-S 1193, Université Paris-Saclay, 94800 Villejuif, France; 3Inserm, Physiopathogénèse et Traitement des Maladies du Foie, Université Paris-Saclay, 94800 Villejuif, France; 4FHU Hepatinov, 94805 Villejuif, France; 5Centre Hospitalier de Hyères, 83400 Hyères, France; 6Groupe Hospitalier Intercommunal Le Raincy-Montfermeil, 78515 Le Raincy, France; 7Centre Hospitalier Annecy Genevois, 74000 Annecy, France; 8Centre Hospitalier Marne-La-Vallée, 77420 Marne La Vallee, France; 9Centre Hospitalier Departemental Vendée, 85000 La Roche sur Yon, France; 10Centre Hospitalier de Meaux, 77100 Meaux, France; 11Centre Hospitalier Sud Francilien, 91100 Corbeil Essonne, France; 12Institut Mutualiste Montsouris, 75014 Paris, France; 13Centre Hospitalier de Dunkerke, 59140 Dunkerke, France; 14Centre Hospitalier Eure-Seine, 27000 Evreux, France; 15Centre Hospitalier Béziers, 34032 Béziers, France; 16Centre Hospitalier Intercommunal de Poissy-St-Germain-en-Laye, 78100 St-Germain-en-Laye, France; 17Centre Hospitalier Bretagne-Sud, 56100 Lorient, France; 18Centre Hospitalier Côte Basque, 64100 Bayonne, France; 19Centre Hospitalier de Perpignan, 66000 Perpignan, France; 20Centre Hospitalier Bry-Sur-Marne, 94360 Bry sur Marne, France; 21Centre Hospitalier Saint-Denis, 93205 Saint Denis, France; 22Clinical Research Unit, Université Paris-Sud, Université Paris-Saclay, 94800 Villejuif, France; 23Département de Virologie, AP-HP Hôpital Paul-Brousse, 94800 Villejuif, France

**Keywords:** hepatitis E, intestinal bowel disease, immunomodulators

## Abstract

Background: Medical treatment of inflammatory bowel disease (IBD) has evolved significantly, and treatment with immunomodulators is recommended. These medications may alter the patient’s immune response and increase the risk of opportunistic infections. Our aim was to evaluate the prevalence and the incidence of acute or chronic HEV infection in IBD patients under immunomodulatory treatment. Patients and Methods: We conducted a retrospective, multicenter, observational study between 2017 and 2018. IBD outpatients hospitalized for the infusion of immunomodulators were included in 16 French centers. During their daily hospitalization, blood samples were drawn for HEV serology (IgM and IgG) and HEV RNA detection. Results: A total of 488 patients were included, of which 327 (67%) patients had Crohn’s disease and 161 (33%) ulcerative colitis. HEV IgM was detected in 3 patients, but HEV RNA was undetectable in all patients. The HEV IgG seroprevalence rate was 14.2%. IgG-positive patients were older at sampling (*p* = 0.01) and IBD diagnosis (*p* = 0.03), had higher seafood consumption (*p* = 0.01) and higher doses of azathioprine (*p* = 0.03). Ileal and upper digestive tract involvement was more frequent in IgG-positive patients (*p* = 0.009), and ileocolic involvement was more frequent in IgG-negative patients (*p* = 0.01). Under multivariate analysis, age > 50 years [OR: 2.21 (1.26, to 3.85), *p* = 0.004] was associated with previous HEV infection. Conclusion: Systematic screening for HEV infection is not needed among IBD patients on immunomodulatory medications. However, in the event of abnormal liver test findings, HEV should be part of the classic diagnostic assessment.

## 1. Introduction

The medical treatment of inflammatory bowel disease (IBD) has evolved significantly, and treatment with immunomodulators or biologic therapy, alone or in combination, is recommended [1]. These medications may alter the patient’s immune response and may increase the risk of opportunistic infections [2].

Hepatitis E is an inflammation of the liver caused by the hepatitis E virus (HEV), which is enterically transmitted [3]. The virus has at least 4 different types: genotypes 1, 2, 3 and 4. Genotypes 1 and 2 have been found only in humans. Genotypes 3 and 4 circulate in several animals, including pigs, wild boars and deer, without causing any disease, and they occasionally infect humans. The global distribution of HEV has distinct epidemiological patterns based on ecology and socioeconomic factors.

In low- and middle-income countries, HEV-1 and -2 are shed in the stool of infected persons. Disease presents as large-scale waterborne epidemics, although endemic diseases within these countries can potentially spread through person-to-person contact or fecally contaminated water and foods [4]. Vertical transmission of HEV from infected mother to fetus is also reported and associated with high fetal and perinatal mortality [5].

In Western countries, HEV infection due to zoonotic HEV-3 and HEV-4 is triggered by infection with virus originating in animals, usually through ingestion of undercooked animal meat (including animal liver, particularly pork). The infection is frequent and mostly asymptomatic or paucisymptomatic [6,7]. Occasionally, a serious disease known as fulminant hepatitis (acute liver failure) develops, which can be fatal [8]. Progression to chronic infection is observed in immunosuppressed patients, such as transplant patients, HIV-infected patients with low CD4 count and patients with hematological diseases treated by chemotherapy [6,7,9].

IBD patients are considered as immunosuppressed since they are mostly treated by immunomodulators. Abnormal liver enzymes are frequently encountered in IBD patients [10,11] and are mostly attributed to nonalcoholic fatty liver disease, primary sclerosing cholangitis (PSC) or drug-induced liver injury (DILI) [12].

To date, there is no prospective study evaluating the prevalence of acute or chronic HEV infection in IBD patients and, therefore, no recommendations regarding whether HEV testing is needed before immunosuppressant treatment is initiated.

Consequently, our aim was to evaluate the prevalence and incidence of acute or chronic HEV infections in patients with IBD under treatment with immunomodulatory medications. Factors associated with HEV seroprevalence were also investigated.

## 2. Materials and Methods

This is a retrospective, multicenter, observational study with prospectively collected data between October 2017 and January 2018. During this period, patients over 18 years old with histologically proven IBD, attending the out-clinic patients of 16 French general hospitals were included consecutively after informed consent. All patients were hospitalized for anti-TNF alpha infusion. During hospitalization, the following data were collected and analyzed: demographics, lifestyle characteristics (urban or rural, North or South France living and alcohol and tobacco consumption), comorbidities (diabetes mellitus, medical history of pulmonary disease and neurologic disorders), HEV risk factors (previous trips, previous transfusions and consumption of seafood, wild boar, deer, pork and figatelli), variables related to IBD (type of IBD, localization, phenotype, age of diagnosis, immunomodulating treatment, duration of treatment, dose of treatment, extra intestinal manifestations and presence of cancer), as well as biological tests (hemoglobin, white blood cells, neutrophil counts, platelet counts, C reactive protein, serum albumin, serum bilirubin, ALT, aspartate transaminase (AST), gamma glutamyl transferase (GGT) and alkaline phosphatase (ALP)) and serum sampling for HEV testing. Patients with a chronic liver disease were not excluded. The cause of the liver disease was made at the investigators’ discretion.

Anti-HEV IgG and IgM were detected by using the Wantai HEV IgG EIA and Wantai HEV IgM EIA kits (Wantai Biologic Pharmacy Enterprise, Beijing, People’s Republic of China), according to the manufacturer’s instructions. HEV RNA was detected from 200 µL of serum, by using the RealStar HEV RT-PCR Kit 1.0 (Altona Diagnostics, Hamburg, Germany), with a lower limit of quantification of 100 IU/mL. Detection of HEV RNA and/or HEV IgM confirmed acute infection. In the case of HEV RNA detection, clinical and biological follow-up was planned in the protocol to identify chronic infection, defined as HEV RNA detection in serum for more than 3 months. The study was approved by the ethics committee of the Bicêtre hospital (Comité de Protection des Personnes Ile-de-France VII) with PP 16-837 reference.

Continuous variables were expressed as means (+/− standard deviations), while qualitative variables were expressed as numbers (percentages). We analyzed the data using SAS/STAT^®^ software. Under univariate analysis, Chi2 and Fisher exact tests were used to compare positive rates of anti-HEV IgG. Student’s t-test or the Mann–Whitney test were used to compare quantitative data. Independent risk factors for HEV IgG positivity (*p*-value < 0.05) were identified by logistic regression analysis among variables selected with a *p*-value < 0.20 from univariate analysis.

## 3. Results

Between October 2017 and January 2018, 488 patients were included—248 (50.8%) were men, and 240 (49.2%) were women. Their mean age was 41.7 ± 14.7 years. The mean time elapsed since the diagnosis of IBD was 10.7 ± 8.1 years. A total of 314 patients (64.3%) lived in northern France and 174 (35.7%) in the south. The characteristics of these patients are described in Table 1.

In total, 327 patients (67%) were suffering from Crohn’s disease (CD), with ileo-colic involvement in 48.5% of cases. A total of 161 patients (33.0%) had ulcerative colitis (UC), which was extensive in 52.4% of cases. IBD was in remission in 421 patients (86.3%). Extra-intestinal manifestations were noted in 109 patients (22.4%). All patients received monoclonal antibodies to maintain remission for a median period of 49.1 ± 38.7 months. These monoclonal antibodies were infliximab (86.7%), vedolizumab (12.5%) and ustekinumab (0.8%). In total, 111 patients (22.8%) received additional treatment with azathioprine, 31 (6.4%) with corticosteroids, 31 (6.4%) with 5-ASA and 12 (2.5%) with budesonide.

Liver enzymes abnormalities were noted for 45 patients (9.2%). The mean AST level was 21.4 ± 9.5 IU/L, and the mean ALT level was 24.2 ± 18.6 IU/L. In 24.4% of these cases, the patients had non-alcoholic steatohepatitis (NASH) associated with metabolic syndrome. The declared diagnosis to explain liver test abnormalities was as follows: PSC for 3 (6.7%) patients, DILI for 3 (6.7%) patients, alcohol liver disease for 2 (4.4%) patients, autoimmune hepatitis for 1 (2.2%) patient and HBV infection for 1 (2.2%) patient. Liver test abnormalities remained unexplained in 22 (48.9%) patients. No patient with acute HEV infection defined by positive IgM anti-HEV had abnormal liver tests.

A total of 69 patients (14.2%) tested positive for anti-HEV IgG. Low levels of HEV IgM were detected in 3 patients (0.6%), but HEV RNA was undetectable in all 488 patients. Since no HEV RNA was detected during follow-up, no cases of chronic HEV infection were detected.

The results of univariate analysis are shown in Table 1. IgG-positive patients were older than IgG-negative patients at inclusion (47.5 years versus 40.1 years, *p* = 0.01), and they were also older at the time of IBD diagnosis, with 34% IgG-positive patients over 40 years versus 22% IgG-negative (*p* = 0.03). Regular seafood consumption was more common among IgG-positive patients (55.2% versus 37.8%, *p* = 0.01). Ileal and upper digestive tract involvement was more frequent in IgG-positive patients (32.8% versus 19%, *p* = 0.009), and ileocolic involvement was more frequent in IgG-negative patients (34% versus 23.9%, *p* = 0.01). Extra-intestinal manifestations were less frequent in IgG-positive patients (13.6% versus 23.8%, 0.04). The mean daily doses of azathioprine were higher in IgG-positive patients (161.4mg versus 135.8mg, *p* = 0.03). There was no difference in terms of previous transfusion (11.9% of IgG-positive patients versus 13.1% of IgG-negative patients, *p* = 0.42), game consumption (19.4% of IgG-positive patients versus 27.8% of IgG-negative patients, *p* = 0.51) or pork consumption (11.9% of IgG-positive patients versus 14.5% of IgG-negative patients, *p* = 0.76).

Under multivariate analysis, age > 50 years [OR: 2.21 (1.26, to 3.85), *p* = 0.004] was associated with previous HEV infection.

## 4. Discussion

By using the Wantai IgG assay, our study showed a 14.2% IgG seroprevalence of HEV in a large group of French IBD patients treated with immunomodulators or biologic therapy.

Wide variations in the sensitivity and specificity rates of anti-HEV IgG assays mean that the interpretation of the differences in seroprevalence are reliable only if the same assay is used [13]. Compared to the results on a blood donor population in France (22.4%) [6], also assessed with the Wantai immunoassay, the seroprevalence of HEV in IBD patients seems lower. However, the reported seroprevalence in French blood donors also varied widely depending on location [6]. Of the 16 centers participating in our study, 12 of them were in an area where the seroprevalence in blood donors is 10–20%. Most included IBD patients (80.1%) have their follow-up in areas of relatively low HEV seroprevalence. A 14.1% seroprevalence in these 390 IBD patients was consistent with that observed in blood donors from the same area. The remaining 19.9% of IBD patients lived in areas where the seroprevalence in blood donors was 20–40%, so none came from a highly endemic area. The HEV seroprevalence in these 97 patients with IBD was 14.4%, thus, lower than that of blood donors from the same region, probably due to different dietary habits than the general population.

Comparison with other studies focusing on IBD patients in other countries is difficult due to different seroprevalences in the general population and the use of different HEV IgG assays [14]. Indeed, most non-Wantai assays underestimate HEV seroprevalence [13]. A Spanish study focused on the seroprevalence of HEV in 87 IBD patients found a rate of 1.14% by using the Diapro, Bioprobes assay [15]. By using the recomWell, Mikrogen assay, a German study of IBD patients observed HEV seroprevalence rates of 17.4% in CD patients and 24.7% in UC patients [16]. By using the Diapro, Bioprobes assay, a Portuguese study of 62 patients with IBD showed an even higher IgG HEV seroprevalence rated at 35.5%, which increased with the degree of immunosuppression [17]. Differences could be (at least in part) due to the serological kits used in the various studies, since the use of different brands is known to yield different results. The sensitivity and specificity of the assays varied widely, making comparisons difficult and highlighting the need to develop a gold standard assay [14,18].

While acute HEV infection has been reported among patients with inflammatory arthritis treated with immunomodulatory medications [19], acute HEV infection appears to be a rare cause of elevated liver enzymes in IBD patients [20]. A French prospective study of 23 patients with rheumatoid arthritis, axial spondylarthritis or other types of arthritis treated using immunomodulatory drugs showed that acute HEV infection was a frequent cause of elevated liver enzymes. Diagnosis of the HEV infection was based on positive PCR results for HEV RNA (n = 14 patients) or anti-HEV IgM positivity (n = 9) [19]. Similar results were produced by other large European cohorts [21]. In our prospective series of a large IBD population treated with immunosuppressive drugs, no acute viremic HEV infections were found, and HEV did not account for unexplained liver enzyme elevations or DILI. Similarly, HEV infection was not found to contribute significantly to the hepatic decompensation among 181 consecutive cirrhosis patients [22]. However, case reports have been published on HEV as a cause of cirrhotic decompensation [23], but it might be too infrequent to be detected in a prospective series.

Another important result of our study was the absence of chronic hepatitis in our population of 488 IBD patients. These negative results suggest that IBD patients treated with immunomodulatory medications are not at risk of developing chronic hepatitis E.

## 5. Conclusions

In conclusion, acute or chronic HEV infection is uncommon in IBD patients who are being treated with immunomodulatory medications. HEV IgG seroprevalence is low in this population and especially older IBD patients. Having previous HEV infection may be related to different dietary habits linked to later and less symptomatic IBD manifestations. Among IBD patients on immunomodulatory medications, systematic screening for HEV infection should not be recommended. However, in the event of abnormal liver test findings, HEV should form part of the diagnostic assessment, in the same way as for any hepatitis, although our study demonstrated a lower exposure to HEV in IBD patients.

## Figures and Tables

**Table 1 pathogens-12-00332-t001:** Characteristics of the population and comparison of IgG-positive versus IgG-negative patients.

Variable *	Overall Population (488 Patients)	HEV IgG-Negative (419 Patients)	HEV IgG-Positive (69 Patients)	*p*-Value
Demographics				
Male/ Female	248 (50.8) /240 (49.2%)	207 (49.2%) /214 (50.8%)	41 (61.2%) /26 (38.8%)	0.07
Age at inclusion	41.7 ± 14.7	40.1 ± 14.9	47.5 ± 15.4	0.01
South France residency	174 (35.7%)	147 (34.9%)	27 (40.3%)	0.39
Risk factors for HEV exposure				
Resident in southern France	174 (35.7%)	147 (34.9%)	27 (40.3%)	0.39
Previous transfusion	56 (11.7%)	50 (11.9%)	6 (8.9%)	0.42
Rural living conditions	193 (39.6%)	167 (39.7%)	26 (38.8%)	0.89
Bore water consumption	44 (9%)	37 (8.8%)	7 (10.4%)	0.41
Raw seafood consumption	196 (40.2%)	159 (37.8%)	37 (55.2%)	0.01
Game consumption	130 (26.6%)	117 (27.8%)	13 (19.4%)	0.51
Pork consumption	69 (14.2%)	61 (14.5%)	8 (11.9%)	0.76
Pork liver consumption	69 (14.2%)	65 (15.4%)	4 (5.8%)	0.79
Liver tests				
Bilirubin (mg/dL)	8.3 ± 4.6	8.41 ± 4.65	7.83 ± 4.35	0.38
AST (IU/L)	21.4 ± 9.5	21.33 ± 9.83	22.06 ± 7.30	0.12
ALT (IU/L)	24.2 ± 18.6	24.14 ± 19.13	24.70 ± 15.28	0.19
GGT (IU/L)	31.8 ± 33.7	31.32 ± 33.54	34.88 ±35.02	0.17
ALP (IU/L)	71.1 ± 25.1	70.35 ± 25.30	76.21 ± 23.41	0.04
IBD characteristics				
Crohn’s disease/ Ulcerative colitis	327 (67%)/ 161 (33%)	281 (66.75%)/ 140 (33.25%)	46 (68.66%)/ 21 (31.34%)	0.76
Ileocolic involvement	159 (48.5%)	143 (34%)	16 (23.9%)	0.01
Ileal and upper digestive tract involvement	17 (5.2%)	80 (19%)	22 (32.8%)	0.009
Remission	421 (86.3%)	365 (86.7%)	56 (83.58%)	0.49
Time since IBD diagnosis	10.7 ± 8.1	10.5 ± 7.8	11.7 ± 9.2	0.57
Age > 40 years at diagnosis	116 (23.8%)	93 (22.1%)	23 (34.3%)	0.03
Extra-intestinal manifestations	109 (22.4%)	100 (23.8%)	9 (13.6%)	0.04
Infliximab treatment	423 (86.7%)	363 (86.22%)	60 (89.55%)	0.48
Vedolizumab treatment	61 (12.5%)	55 (13.06%)	6 (8.96%)	0.48
Ustekinumab treatment	4 (0.8%)	3 (0.71%)	1 (1.49%)	0.48
Duration of monoclonal antibody treatment (years)	4.1 ± 3.2	7.9 ± 6.3	6.7 ± 4.9	0.27
Azathioprine	111(22.8%)	100 (3.75%)	11 (16.32%)	0.36
Azathioprine daily dose (mg)	138.3 ± 40.4	135.8 ± 40.4	161.4 ± 34.2	0.03
Corticosteroids ongoing treatment	31 (6.4%)	26 (6.18%)	5 (7.46%)	0.67

* Continuous variables are expressed as mean (+/− interquartile). Qualitative variables are expressed as numbers (percentages). HEV: Hepatitis E virus, AST: aspartate aminotransferase, ALT: alanine aminotransferase, GGT: gamma-glutamyl transferase, ALP: Alkaline phosphatase, IBD: Inflammatory bowel disease.

## Data Availability

The data underlying this article will be shared after request to the corresponding author.

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
