# Peer review of "Hepatitis E Virus Infection in Patients with Chronic Inflammatory Bowel Disease Treated with Immunosuppressive Therapy"

_pathogens, 2023, doi:10.3390/pathogens12020332_

Round 1

Reviewer 1 Report

The study performed by Kounis et al. evaluated HEV infection in patients with inflammatory bowel disease (IBD), a group that has been little studied regarding infection with HEV. It is well written, short and to the point, and a relatively large number of individuals were included in the study, which enriches it and further supports the results.

Some corrections are needed:

-Lines 102 and 103: change “Mean AST levels was 21.4 ± 9.5 IU/l and mean ALT levels was 24.2 ± 18.6 IU/l” by “Mean AST level was 21.4 ± 9.5 IU/l and mean ALT level was 24.2 ± 18.6 IU/l.

-In line 116 the authors stated that no chronic hepatitis E cases were detected, but no definition of chronic hepatitis E was given (only acute HEV infection was defined). Please, clarify this in the Materials and Methods section (acute infection: detection of HEV RNA in one serum and/or stool sample. Chronic HEV infection: detection of HEV RNA in serum and/or stool samples for more than 3 months).

-The Discussion section begins: “Our study showed a low prevalence (0.6%) of HEV in a large group of IBD patients treated with immunomodulators or biologic therapy”. However, the value 0.6% is not the prevalence, since it is the percentage of IgM anti-HEV detection, previously defined as incidence (not prevalence, line 116). On the other hand, the cited studies in the first paragraph of the Discussion (from Spain, Germany, etc.), refer to IgG prevalence values, so the comparisons made are not correct (IgM detection percentage is compared with IgG detection percentage). Please, rewrite this paragraph with the value of the HEV IgG prevalence in the cohort of this study (13.7%).

-Line 150: the word “in” is written twice. Please, delete one.

-In the Discussion section, some explanations are given for the differences in the HEV prevalences obtained in the study with respect to other studies performed in IBD patients and with respect to blood donors from France. Although diet could be one of the explanations, it is very important to mention that the differences could be (at least in part) due to the serological kits used in the various studies, since the use of different brands is known to yield different results (see articles: Pisano et al. 2022, Pisano et al. 2018). This should be stated in the Discussion.

Articles:

Pisano et al. 2022. Hepatitis E virus infection in the United States: seroprevalence, risk factors and the influence of immunological assays. Plos One. 2022. 17(8):e0272809. doi: 10.1371/journal.pone.0272809.

Pisano et al. 2018. Hepatitis E virus in South America: the current scenario. Liver International. 2018. 38 (9): 1536-1546. doi: 10.1111/liv.13881.  

Another possible reason for the difference in the HEV prevalences obtained in this study and that reported for the blood bank from France, is the geographic region where the samples were obtained, since differences in prevalences have been described according to the geographic region, even within the same country. Please, check this information and clarify this in the Discussion section (in addition to the used kit to determine IgG anti-HEV).

-Conclusion, lines 174-175: it is stated that “HEV prevalence is low in this population and at its lowest levels among patients whose IBD is severe and was of early onset”. What do you mean by “its lowest levels among patients whose IBD is severe and was of early onset”? This is not clear.

Author Response

Point by point responses to reviewers

REVIEWER 1

The study performed by Kounis et al. evaluated HEV infection in patients with inflammatory bowel disease (IBD), a group that has been little studied regarding infection with HEV. It is well written, short and to the point, and a relatively large number of individuals were included in the study, which enriches it and further supports the results.

Some corrections are needed:

  • Lines 102 and 103: change “Mean AST levels was 21.4 ± 9.5 IU/l and mean ALT levels was 24.2 ± 18.6 IU/l” by “Mean AST level was 21.4 ± 9.5 IU/l and mean ALT level was 24.2 ± 18.6 IU/l.

Thank you for your comment. Please find the correction in lines 102-103.

  • In line 116 the authors stated that no chronic hepatitis E cases were detected, but no definition of chronic hepatitis E was given (only acute HEV infection was defined). Please, clarify this in the Materials and Methods section (acute infection: detection of HEV RNA in one serum and/or stool sample. Chronic HEV infection: detection of HEV RNA in serum and/or stool samples for more than 3 months).

Thank you for your comment. In our protocol we considered acute HEV when HEV RNA and/or HEV IgM were detected and chronic HEV infection when HEV RNA was detected in serum for more than 3 months. In case of HEV RNA detection at baseline, we planned clinical and biological follow up to identify chronic infection. Please find the corrections in lines 112-115.

  • The Discussion section begins: “Our study showed a low prevalence (0.6%) of HEV in a large group of IBD patients treated with immunomodulators or biologic therapy”. However, the value 0.6% is not the prevalence, since it is the percentage of IgM anti-HEV detection, previously defined as incidence (not prevalence, line 116).

Thank you for your comment. Indeed, the value 0.6% is the percentage of IgM anti HEV detection which we defined as acute HEV infection. Please find the correction in line 116.

  • On the other hand, the cited studies in the first paragraph of the Discussion (from Spain, Germany, etc.), refer to IgG prevalence values, so the comparisons made are not correct (IgM detection percentage is compared with IgG detection percentage). Please, rewrite this paragraph with the value of the HEV IgG prevalence in the cohort of this study (13.7%).

Thank you for your comment. We have clarified this important subject in the discussion section and compared the IgG seroprevalence in the different countries (lines 192-204). Comparison with other studies focusing on IBD patients in other countries is difficult due to different seroprevalences in the general population and the use of different HEV IgG assays. The Spanish study found a rate of 1.14% IgG seroprevalence by using the Diapro, Bioprobes assay. The German study observed HEV prevalence rates of 17.4% in CD patients and 24.7% in UC patients by using the recomWell, Mikrogen. Finally the Portuguese study showed a IgG prevalence of  35.5% by using the Diapro, Bioprobes assay.

  • Line 150: the word “in” is written twice. Please, delete one.

Thank you for your comment. Please find the correction in line 151.

  • In the Discussion section, some explanations are given for the differences in the HEV prevalences obtained in the study with respect to other studies performed in IBD patients and with respect to blood donors from France. Although diet could be one of the explanations, it is very important to mention that the differences could be (at least in part) due to the serological kits used in the various studies, since the use of different brands is known to yield different results (see articles: Pisano et al. 2022, Pisano et al. 2018). This should be stated in the Discussion.

Thank you for your remark. Indeed, differences could be related to different serological kits. However, the French study in liver donors used also Wantai IgG assay, like our study, and found a higher prevalence of IgG seroprevalence (22.4%). In addition to different dietary habits, reported seroprevalence in French blood donors may also have varied because of differences in locations. Of the 16 centers participating in our study, 12 of them were located in an area where seroprevalence in blood donors is of 10-20%. Most included IBD patients (80.1%) have their follow-up in areas of relatively low HEV seroprevalence. A 14.1% seroprevalence in these 390 IBD patients was consistent with that observed in blood donors from the same area. The remaining 19.9% of IBD patients lived in areas where seroprevalence in blood donors was 20-40%, so none came from a highly endemic area. HEV seroprevalence in these 97 patients with IBD was 14.4%, thus lower than that of blood donors from the same region. Please find the revised version in lines 178-191. 

  • Conclusion, lines 174-175: it is stated that “HEV prevalence is low in this population and at its lowest levels among patients whose IBD is severe and was of early onset”. What do you mean by “its lowest levels among patients whose IBD is severe and was of early onset”? This is not clear.

Thank you for your comment. We rephrased in lines 225-226 writing that HEV IgG seroprevalence is low in this population and especially in older IBD patients. This result comes from the fact that age>50 years [OR: 2.21 (1.26, to 3.85), p=0.004] was associated with previous HEV infection in multivariate analysis.

Reviewer 2 Report

The authors have tested 488 patients with Inflammatory bowel disease on immunomodulators for markers of HEV infection (IgM anti-HEV & HEV RNA) and seroprevalence (IgG anti-HEV). 67 patients were reactive for IgG anti-HEV, 3 for IgM anti-HEV, and none for HEV RNA. I have the following comments: (i) Introduction of the manuscript is very short and does not define why this study is important. Important features of HEV- namely HEV gt1 & pregnancy (Viruses 202113(7), 1329; https://doi.org/10.3390/v13071329) and HEV-gt3 in those with SOT, hemopoietic NP, and HIV  ( Clin Microbiol Rev. 2014 Jan;27(1):116-38. doi: 10.1128/CMR.00057-13) needs to be highlighted. (ii) Same paragraph should mention the mode of transmission of HEV in East and West and the role of Figatelli sausage and other swine products in the spread of HEV in Europe (Viruses 20168(9),253; https://doi.org/10.3390/ v8090 253).  (ii) It is not clear from the paper how many times each patient was tested for markers of HEV. Was it once or each time patients attended the clinic for immunomodulators between 2017-2018? ((iii) Did any of the patients (488) suffer from acute hepatitis syndrome or chronic hepatitis or cirrhosis and if so what was the etiology? (iv) Of the 67 patients with IgG anti-HEV and 3 with IgM anti-HEV did any patient have elevated liver enzymes?  

Author Response

Point by point responses to reviewers

REVIEWER 2

The authors have tested 488 patients with Inflammatory bowel disease on immunomodulators for markers of HEV infection (IgM anti-HEV & HEV RNA) and seroprevalence (IgG anti-HEV). 67 patients were reactive for IgG anti-HEV, 3 for IgM anti-HEV, and none for HEV RNA. I have the following comments:

  • Introduction of the manuscript is very short and does not define why this study is important. Important features of HEV- namely HEV gt1 & pregnancy (Viruses202113(7), 1329; https://doi.org/10.3390/v13071329) and HEV-gt3 in those with SOT, hemopoietic NP, and HIV  ( Clin Microbiol Rev. 2014 Jan;27(1):116-38. doi: 10.1128/CMR.00057-13) needs to be highlighted. Same paragraph should mention the mode of transmission of HEV in East and West and the role of Figatelli sausage and other swine products in the spread of HEV in Europe (Viruses 20168(9),253; https://doi.org/10.3390/ v8090 253). 

Thank you for your comment that we take into consideration. Please find in lines 58-81 a revised introduction featuring the HEV genotypes and the different ways of transmission of the virus.

  • It is not clear from the paper how many times each patient was tested for markers of HEV. Was it once or each time patients attended the clinic for immunomodulators between 2017-2018?

In our protocol patients were tested once at baseline and we considered acute HEV when HEV RNA and/or HEV IgM were detected. In case of detection of HEV RNA detection at inclusion, we had planned a clinical and biological follow up to identify a chronic infection, defined as HEV RNA detected in serum for more than 3 months. Please find the corrections in lines 112-115.

  • Did any of the patients (488) suffer from acute hepatitis syndrome or chronic hepatitis or cirrhosis and if so what was the etiology? Of the 67 patients with IgG anti-HEV and 3 with IgM anti-HEV did any patient have elevated liver enzymes?  

Thank you for your excellent comment. No patient with acute HEV infection defined by positive IgM anti HEV had abnormal liver tests. No patient developed chronic hepatitis or cirrhosis related to HEV. This is the reason why we concluded that among IBD patients on immunomodulatory medications, systematic screening for HEV infection should not be recommended when liver tests are normal. Forty-five patients (9.2%) had abnormal liver tests, with a mean AST level at 21.4 ± 9.5 IU/l and a mean ALT level at 24.2 ± 18.6 IU/l. The declared diagnosis to explain liver test abnormalities were as follow: non-alcoholic steatohepatitis (NASH) for 11 (24.4%) patients, PSC for 3 (6.7%) patients, DILI for 3 (6.7%) patients, alcohol liver disease for 2 (4.4%) patients, autoimmune hepatitis for 1 (2.2%) patient and HBV infection for 1 (2.2%) patient. Liver test abnormalities remained unexplained in 22 (48.9%) of patients.

Reviewer 3 Report

Kounis I, et al. demonstrated the seroprevalence of HEV in patients with chronic inflammatory bowel diseaseses treated with immunosuppressive therapy and found low positive rates of HEV infection, using relative large sample-size. This paper contains interesting and valuable information but has several issues to be addressed for publication.

Major

1.       Authors compared seropositivities of IgG or IgM antibodies against HEV to other reports, using different ELISA kits. These comparison cannot be easily done otherwise authors check these ELISA kits, using same samples? Different ELISA kits sometimes show different results.

2.       The seropositive rates were too low, compared with the nation-wide surveillance in France (Ref #3) although they used the same ELISA kits. Authors concluded this this difference as different dietary habits, but considerable patients had risk food for HEV. Therefore, I am wondering the possibilities that immunosuppressive treatment impaired antibody production.

3.       Have ever checked seropositivities for HEV in patients with inflammatory bowel diseases without immunosuppressive therapies.

Author Response

Point by point responses to reviewers

REVIEWER 3

  • Authors compared seropositivities of IgG or IgM antibodies against HEV to other reports, using different ELISA kits. These comparison cannot be easily done otherwise authors check these ELISA kits, using same samples? Different ELISA kits sometimes show different results.

Thank you for your excellent remark. Comparison with other studies focusing on IBD patients in other countries is difficult due to different HEV IgG assays since most non-Wantai assays underestimate HEV seroprevalence. The Spanish study found a rate of 1.14% IgG seroprevalence by using the Diapro, Bioprobes assay. The German study observed HEV prevalence rates of 17.4% in CD patients and 24.7% in UC patients by using the recomWell, Mikrogen. The Portuguese study showed a IgG prevalence of  35.5% by using the Diapro, Bioprobes assay. Finally, the French study in liver donors used also Wantai IgG assay, like our study, and found a higher prevalence in blood donors (22.4%) than the population of IBD patients. Please find the revised version in lines 178-204 of the discussion section.

  • The seropositive rates were too low, compared with the nation-wide surveillance in France (Ref #3) although they used the same ELISA kits. Authors concluded this this difference as different dietary habits, but considerable patients had risk food for HEV. Therefore, I am wondering the possibilities that immunosuppressive treatment impaired antibody production.

Thank you for your remark. Although comparison of seroprevalence may be difficult because of the use of different serological kits, the French study in liver donors used the same kit as our study (Wantai IgG assay). A possible explanation might be different dietary habits and also, as you suggest, impaired antibody production in case of immunosuppressive treatment. However we suggest that differences may be related to variations in locations. Of the 16 centers participating in our study, 12 of them were located in an area where seroprevalence in blood donors is of 10-20%. Most included IBD patients (80.1%) have their follow-up in areas of relatively low HEV seroprevalence. A 14.1% seroseroprevalence in these 390 IBD patients was consistent with that observed in blood donors from the same area. The remaining 19.9% of IBD patients lived in areas where seroprevalence in blood donors was 20-40%, so none came from a highly endemic area. HEV seroprevalence in these 97 patients with IBD was 14.4%, thus lower than that of blood donors from the same region. Please find the revised version in lines 178-191. 

  • Have ever checked seropositivities for HEV in patients with inflammatory bowel diseases without immunosuppressive therapies.

Thank you for your remark. Unfortunately, we did not include a control group of IBD patients with no immunomodulatory treatment. All the patients included had anti TNF treatment. However, 5 (7.5%) IgG positive patients had a combination treatment with steroids, 1 (1.5%) with methotrexate and 11 (16.3%) with azathioprine.

Reviewer 4 Report

The article “Hepatitis E virus infection in patients with chronic inflammatory bowel disease treated with immunosuppressive therapy” describes an important problem of HEV infection, and its relevance is beyond doubt. The paper starts with an extremely short introduction; it should be expanded. The methods described are adequate, and the results of the paper are well discussed. Here are a few comments and suggestions for authors:

1.       The authors write that immunomodulators or biological therapy might influence the prevalence of HEV infection. Also, the medications may alert the immune response and increase the risk of infections, including HEV infection. More Information on HEV or other virus examples in the Introduction section should be added.

2.       In the abstract, the authors write, “HEV IgM was detected in 3 patients, but HEV RNA was undetectable in all patients; the HEV seroprevalence rate was 13.7%.”. In the Results section, “A total of 67 patients (13.7%) tested positive for anti-HEV IgG. Low levels of HEV IgM were detected in 3 patients (0.6%), but HEV RNA was undetectable in all 488 patients. The incidence of HEV was therefore 0.6%, and no cases of chronic HEV infection were detected.” Those 3 patients were IgM and IgG positive? I think there is a mistake in calculating the rate of HEV seroprevalence. In my opinion, it should be 14.3 %. Or I do not understand.

3.       The detection of HEV RNA confirms the incidence of HEV, not specific IgM. The sentences (Line 116) “The incidence of HEV was therefore 0.6% and no cases of chronic HEV infection were detected.” and (Lines 141,142) “Our study showed a low prevalence (0.6%) of HEV in a large group of IBD patients treated with immunomodulators or biologic therapy.”  should be rewritten/corrected.

4.       Line 165. The abbreviation for DILI should be explained.

5.       Line 87. The sentence “Acute HEV was considered when HEV RNA was detected.” might be replaced with “Detection of HEV RNA confirms acute and chronic HEV infection.”

6.       Information about the sensitivity and specificity of the used commercial serological and molecular tests should be added. 

Author Response

Point by point responses to reviewers

REVIEWER 4

The article “Hepatitis E virus infection in patients with chronic inflammatory bowel disease treated with immunosuppressive therapy” describes an important problem of HEV infection, and its relevance is beyond doubt. The paper starts with an extremely short introduction; it should be expanded. The methods described are adequate, and the results of the paper are well discussed. Here are a few comments and suggestions for authors:

  • The authors write that immunomodulators or biological therapy might influence the prevalence of HEV infection. Also, the medications may alert the immune response and increase the risk of infections, including HEV infection. More Information on HEV or other virus examples in the Introduction section should be added.

Thank you for your comment that we take into consideration. Please find in lines 58-81 a revised introduction section.

  • In the abstract, the authors write, “HEV IgM was detected in 3 patients, but HEV RNA was undetectable in all patients; the HEV seroprevalence rate was 13.7%.”. In the Results section, “A total of 67 patients (13.7%) tested positive for anti-HEV IgG. Low levels of HEV IgM were detected in 3 patients (0.6%), but HEV RNA was undetectable in all 488 patients. The incidence of HEV was therefore 0.6%, and no cases of chronic HEV infection were detected.” Those 3 patients were IgM and IgG positive? I think there is a mistake in calculating the rate of HEV seroprevalence. In my opinion, it should be 14.3 %. Or I do not understand.

Thank you for your remark. You are absolutely right. We corrected this mistake in the new version of the manuscript in lines 147 and 175.

  • The detection of HEV RNA confirms the incidence of HEV, not specific IgM. The sentences (Line 116) “The incidence of HEV was therefore 0.6% and no cases of chronic HEV infection were detected.” and (Lines 141,142) “Our study showed a low prevalence (0.6%) of HEV in a large group of IBD patients treated with immunomodulators or biologic therapy.”  should be rewritten/corrected.

Thank you for your comment. Indeed, the value 0.6% is the percentage of IgM anti HEV detection which we defined as acute HEV infection. Please find the correction in line 116.

  • Line 165. The abbreviation for DILI should be explained.

Thank you for your comment. Please find the abbreviation in line 81.

  • Line 87. The sentence “Acute HEV was considered when HEV RNA was detected.” might be replaced with “Detection of HEV RNA confirms acute and chronic HEV infection.”

Thank you. We took into consideration your comment and revised in line 111. 

Round 2

Reviewer 2 Report

The authors have addressed all issues raised in my review 1. 

Reviewer 3 Report

Authors correctively revised.